# Wider and Stronger Inhibitory Ring of the Attentional Focus in Schizophrenia

**DOI:** 10.3390/brainsci13020211

**Published:** 2023-01-27

**Authors:** Luca Ronconi, Vincenzo Florio, Silvia Bronzoni, Beatrice Salvetti, Agnese Raponi, Giancarlo Giupponi, Andreas Conca, Demis Basso

**Affiliations:** 1School of Psychology, Vita-Salute San Raffaele University, 20132 Milan, Italy; 2Division of Neuroscience, IRCCS San Raffaele Scientific Institute, 20132 Milan, Italy; 3Psychiatric Service of the Health District of Bozen, 39100 Bozen, Italy; 4CESLab, Faculty of Education, Free University of Bozen, 39042 Brixen, Italy; 5Centro de Investigación en Neuropsicologia y Neurociencias Cognitivas (CINPSI Neurocog), Universidad Católica del Maule, Av. San Miguel, Talca 3480094, Chile

**Keywords:** visual attention, Mexican hat, surround suppression, spatial inhibition, psychotic disorders

## Abstract

Anomalies of attentional selection have been repeatedly described in individuals with schizophrenia spectrum disorders. However, a precise analysis of their ability to inhibit irrelevant visual information during attentional selection is not documented. Recent behavioral as well as neurophysiological and computational evidence showed that attentional search among different competing stimuli elicits an area of suppression in the immediate surrounding of the attentional focus. In the present study, the strength and spatial extension of this surround suppression were tested in individuals with schizophrenia and neurotypical controls. Participants were asked to report the orientation of a visual “pop-out” target, which appeared in different positions within a peripheral array of non-target stimuli. In half of the trials, after the target appeared, a probe circle circumscribed a non-target stimulus at various target-to-probe distances; in this case, participants were asked to report the probe orientation instead. Results suggest that, as compared to neurotypical controls, individuals with schizophrenia showed stronger and spatially more extended filtering of visual information in the areas surrounding their attentional focus. This increased filtering of visual information outside the focus of attention might potentially hamper their ability to integrate different elements into coherent percepts and influence higher order behavioral, affective, and cognitive domains.

## 1. Introduction

The study of how humans select relevant visual information and inhibit irrelevant information is of high relevance for understanding sensory and cognitive profiles in individuals with schizophrenia spectrum disorders (SSD) [1,2].

Traditional models conceptualized spatial attention through the metaphors of a “spotlight” or a “zoom-lens”. According to the spotlight model, a fixed-size lens can be oriented to a particular location in the visual field to better extract relevant signals and prepare action (e.g., eye movements) toward a relevant location [3,4,5]. According to the zoom-lens model, the attentional focus can change in its size to adjust the spatial limits of the areas/objects under attentional selection, similar to a “zoom-lens” camera that can shrink or expand the area under its focus [6,7,8,9]. Studies regarding attentional selection in SSD conducted under these traditional frameworks (spotlight and zoom-lens models) have shown that rapid orienting of spatial attention seems to be relatively enhanced in individuals with SSD as compared to neurotypical individuals, especially when using endogenous spatial cues (e.g., arrows), coherently with other enhancements of information processing observed in SSD [10]. Interestingly, some forms of attentional orienting to social cues seem to be altered in SSD, with reports showing both an impaired [11,12] or enhanced [13,14] attentional orienting to gaze cuing. In terms of the spatial scaling of the attentional focus, behavioral and psychophysiological evidence shows a hyperfocusing of attention at the point of gaze and increased filtering of peripheral distractors when the task requires a narrow focusing of attention [15,16]. As an alternative to the hyperfocusing hypothesis, other findings showed that deficits in SSD seem to rely on the precise adjustment of the attentional zoom-lens size [17,18].

Conversely, little research has studied the ability of people with SSD to inhibit irrelevant visual information during attentional selection. Based on more recent psychophysical, neurophysiological and computational evidence, attentional selection, especially when searching among different competing stimuli, seems to be characterized by an area of attenuated excitability in the immediate surround of the attentional focus. This inhibitory annulus gives rise to a center-surround profile, often referred to as the “Mexican hat” profile of the attentional focus [3,19,20,21,22]. Differently from what was predicted by the spotlight and zoom-lens models, attentional resolution would not fade progressively (quasi-linearly) as the distance from the center of the attentional focus increases. Instead, it would be characterized by a non-linear function where resolution is strongly limited in spatial locations nearby the center of the attentional focus, and then partly recovered as the distance from the focus increases. This spatial profile is ideal not only for amplifying relevant target information without simultaneously boosting noise in a visual scene, but also for ensuring proper spatial integration of nearby different visual elements into a coherent percept. Indeed, this exact profile is evident only when stimulus discrimination entails a precise spatial localization [19,20,23]. The computational account ‘selective tuning model’ [24,25] predicted that such center-surround profile is mediated by a recurrent top-down winner-take-all mechanism that starts from the winning units in higher-level visual areas most activated after the initial feed-forward flow and eliminates projections from lower-level units that do not contribute to the attended target location [26].

Evaluating the intensity and extension of this inhibitory ring in SSD would provide important insight into the attentional mechanisms in this clinical condition and the potential link with higher order cognitive dysfunctions. In the present study, we evaluate the spatial profile of the attentional focus in individuals with schizophrenia using a task where visual attention was captured automatically by a pop-out target (red C) embedded in an array of non-target stimuli (blue C). In half of the trials (target condition), participants reported the orientation of the target, which always appears in a different position. In the other half of the trials (probe condition), after the target appearance, a probe circle circumscribed a non-target stimulus at various target-to-probe distances. This latter condition allowed us to measure the spatial profile of the attentional focus, and thus the extension of its inhibitory ring. According to the initial evidence reviewed above suggesting a tendency in SSD to exhibit hyperfocusing of attention and increased filtering of distractors, we hypothesized that the strength of inhibition for spatial positions nearby the attentional focus center should be increased.

## 2. Materials and Methods

### 2.1. Participants

Forty-six adult participants took part initially in the present experiment on a voluntary basis. The patients’ group included 26 participants (15 males) and the control group included 20 participants (11 males). Two participants from the patients group and 2 from the control group were excluded from statistical analyses since their overall accuracy in the target condition (see Procedure) was below the chance level (≤25%) or at ceiling (≥95%). Thus, the final samples comprised 24 participants for the patients’ group and 18 for the control group. The mean age of patients was 34.4 (range: 21–62) and did not differ significantly from the mean age of neurotypical controls, which was 40.25 (range: 19–60) (t_(40)_ = −1.52, *p* = 0.138).

All patients involved in the present study received a clinical diagnosis of schizophrenia and were recruited at the Psychiatric Service of the Health District of Bozen, Bozen, Italy. For all patients, the severity of their condition was stable at the time of the experiment, and they were receiving a stable dose of antipsychotic medications at the time of the assessment (for the details see Table 1). They all had normal/corrected-to-normal vision and hearing. Diagnosis of schizophrenia was made by licensed clinicians in adherence to the DSM-IV-TR criteria using the Structured Clinical Interview (SCID) for DSM-IV-TR. The severity of schizophrenia symptoms was evaluated using the Scale for the Assessment of Negative Symptoms (mean score: 81.8, range: 65–92) and the Scale for the Assessment of Positive Symptoms (mean score: 75.9, range: 60–90). Although, as specified above, all our patients had a diagnosis of schizophrenia, we will use hereafter the label SSD to refer to our group of patients to better adhere to the current standards in the scientific literature.

Participants of the control group were sampled from the same geographical area and were defined as ‘neurotypical’ since they did not have prior history of any neurological and/or psychiatric disorders. Informed consent was obtained from each participant and the entire research protocol was conducted in accordance with the principles elucidated in the declaration of Helsinki. The study was approved by the Ethical Committee of Bolzano Hospital.

### 2.2. Apparatus and Stimuli

The experiment was adapted from a recent study by Ronconi et al. [22] (see Figure 1). Stimulus presentation and data acquisition were achieved with E-Prime 2. Participants seated in a dimly lit and quiet room, at 50 cm of distance from a LCD screen (Hp Compaq 1720, 17 inches, 60 Hz) connected to a laptop. A chin rest was used to avoid head movement.

All stimuli were presented on a middle grey background. The fixation point consisted of a black cross subtending 0.5 deg, presented in the screen center. The search array was displayed randomly in the left or right visual hemifield, and was composed of nine “C-like” (hereafter C for simplicity) stimuli, all initially colored in blue; the target C stimulus was the only one that changed color from blue to red. Both target and non-target Cs subtended a visual angle of 1.2 deg and were presented at an iso-eccentric distance of 8.25 deg from the fixation. All Cs were created from a ring-shaped stimulus by removing a portion subtending a 45° angle; this missing portion varied randomly in position (up, down, left and right) for each stimulus of the array and for each trial. One out of the 9 total Cs was presented aligned with the horizontal axis; the other Cs were presented with four in the upper and four in the lower quadrant, separated by an angle of 0.6 deg edge-to-edge. The stimulus used as a probe was a white circle with a diameter of 2.12 deg. Mask stimuli were obtained from the complete ring-shaped stimuli used to create the C. 

### 2.3. Procedure

Participants were asked to maintain their eyes on the fixation for the entire duration of a trial. In each trial, the fixation cross (1000 ms) anticipated the onset of the stimulus array composed by nine randomly oriented (non-target) blue C in the left or right hemifield. After 50 ms (3 refresh cycles), a target C was colored in red for 100 ms (6 refresh cycles); it could appear randomly in one of the nine possible stimulus locations within the array. In the ‘target’ condition (50% of the trials), after the presentation of the red target C, all stimuli were replaced by masks (duration = ~17 ms, 1 refresh cycle) and then the trial ended. In the ‘probe’ condition (remaining 50% of the trials), the appearance of the target (red C) was followed by the probe circle appearing around the central blue C for 50 ms (3 refresh cycles). In probe trials, the red C remained visible for the entire duration of the probe and subsequently was replaced by masks.

After the mask offset, for both target and probe trials, a blank screen (1000 ms) anticipated the appearance of the response screen containing the four possible orientations of the C (colored in red for target trials and in blue for probe trials). Participants then indicated their assumed correct orientation of the red C in the target condition and of the blue C (highlighted by the probe) in the probe condition. The experimenter entered the selected choice. We specified to participants that only accuracy was evaluated. Considering that the probe position remained constant and the target position varied, there were five target-to-probe distances, hereafter probe distance (PD), ranging PD0 (probe at the target location) to PD4 (probe at the farthest distance). The entire experiment consisted of 144 trials (preceded by 12 practice trials), 72 for the target and 72 for the probe condition, which were randomly selected during the experiment.

The present experimental design was optimal to test the center-surround profile of the attentional focus since attention was first oriented to one of the two hemifields by the appearance of the stimulus array for 50 ms. Then, the focus of attention was captured by the pop-out target for 100 ms, which was followed by the probe for another 50 ms. This sequence of stimuli was over before participants could make an eye-movement towards the peripheral array, while using a timing of events compatible with previous psychophysical studies measuring the surround suppression of the attentional focus [19,22,27,28].

### 2.4. Data Analysis

As a dependent variable to use in the mixed-design repeated measure ANCOVA, we calculated the difference between the accuracy observed in the probe condition (Figure 1B and Figure 2) and the accuracy observed in the target condition (where no probe was presented) (Figure 1A and Figure 2). This measure is supposed to be higher in spatial positions where attentional suppression is stronger, that is, in positions surrounding the center of the attentional focus, and for this reason we will refer to it as *attentional suppression index* (ASI) (Figure 2). The ANCOVA was performed on ASI considering the probe distance (PD; PD0 = center of the attentional focus; PD1 = position adjacent to the attentional focus center; etc.) as within-subjects factor and the group (SSD vs. Controls) as between-subjects factor. Age was inserted as a covariate given the high age range present in our participants, and the Greenhouse-Geisser correction was applied when the sphericity assumption was violated. Planned comparisons were performed using the false discovery rate (FDR) correction for multiple comparisons [29].

## 3. Results

The ANCOVA did not reveal main effects of PD (F_(3.26, 127.18)_ = 1.93, *p* = 0.108), Group (F_(1, 39)_ = 1.74, *p* = 0.194) or Age (F_(1, 39)_ = 0.111, *p* = 0.741). Importantly, a significant PD by Group interaction emerged (F_(4, 156)_ = 2.66, *p* = 0.035; Figure 2C). This interaction was further explored with one-sample *t*-tests against 0 within each group and with independent samples *t*-tests (one tail), which were performed at the different probe distances where surround inhibition should emerge (i.e., PD1–PD4). Regarding one-sample *t*-tests, ASI values equal to 0 represent the absence of attentional suppression for a specific probe distance. In the control group, the ASI was significantly different from 0 only at PD1 (t_(17)_ = 4.76, p_corr_ < 0.001) but not in all other PDs (all ps > 0.116). In the SSD group, the ASI was significantly different from 0 at PD1 (t_(23)_ = 7.51, p_corr_ < 0.001), PD2 (t_(23)_ = 6.23, p_corr_ < 0.001) and PD4 (t_(23)_ = 3.41, p_corr_ = 0.005). Finally, independent samples *t*-tests revealed that the ASI was higher in the SSD group as compared to the control group in PD2 (t_(40)_ = −2.43, p_corr_ = 0.044). Collectively, these results report that the SSD group showed a stronger and wider attentional suppression for information presented outside the attentional focus as compared to the control group.

## 4. Discussion

In the present study, the spatial profile of the attentional focus in individuals with SSD was systematically investigated, to provide insights into the functionality of visual attention in individuals affected by this condition. Inspired by similar studies performed in neurotypical adults that showed a center-surround profile of the attentional focus, the present study presented a unique task able to measure the two key features of visuo-spatial attentional selection in individuals with SSD: (i) the capacity to select information at relevant spatial locations, and (ii) the ability to filter out irrelevant information located at different distances from the focus of attention.

As compared to neurotypical controls, individuals with SSD show stronger filtering of visual information for spatial positions outside their focus of attention. Moreover, they show a wider spatial extension of the inhibitory annulus surrounding the focus of attention. Thus, results observed in the SSD group matched substantially with the hypothesis of an hyperfocusing of attention in SSD, which has been proposed by previous studies [15,16]. Hyperfocusing, indeed, should lead to increased filtering of information falling outside the attentional focus, both in terms of filtering strength and in terms size of the inhibitory areas. Our findings confirmed both these predictions, showing that the filtering of visual information outside the focus of attention is more accentuated and shows a wider spatial extension in individuals with SSD compared to neurotypical individuals. This pattern is particularly evident when looking at the attentional suppression index (ASI) at PD2 in the two groups; indeed, individuals with SSD in this position showed a stronger ASI as compared to controls. Moreover, in this same spatial position, there was a complete recovery from suppression in controls; indeed, the ASI was not significantly different from 0 at PD 2, 3 and 4 in the control group, supporting the claim that the suppressive ring effect was significant only at PD1. Contrarily, the attentional suppression was still evident in the SSD group at PD2 and at PD4.

Such stronger and wider suppression of information surrounding the focus of attention might arise from an unbalanced relationship between neural mechanisms of enhancement and suppression near the locus of visual attention, and it is likely to significantly affect how people with SSD integrate visual information across their visual field. Indeed, in studies on visual perception, SSD showed a poorer ability to integrate different basic visual elements in coherent percepts. This is the case for static elements such as Mooney faces [30,31], and also for dynamic visual displays containing biological motion information [32]. Moreover, patients with SSD have been consistently associated with higher resistance to illusions involving high-level integration processes [33]. Visual illusions are an effective tool to probe visual perceptual functioning in clinical populations. Likewise, starting from this evidence, it is critical to understand the neurocognitive mechanisms behind this atypical integrative processing of illusory and non-illusory images. Our results suggest that one of such mechanisms might be identified in a peculiar spatial pattern of visual selection at the locus of attention, which might globally alter spatial integration of elements across the visual field (for static, dynamic, and illusory visual information).

In our study the deployment of spatial attention and the center-surround mechanisms of the attentional focus was measured within a pre-saccadic time window. Nonetheless, since covert attentional mechanisms (i.e., in the absence of eye movements) are crucial to plan accurately saccadic movements toward the source of relevant information in the environment [34,35,36], such a peculiar spatial profile of attentional focus in SSD might be also linked to different ocular behavior during more ecological visual exploration tasks. For example, differences in eye movements between individuals with SSD and controls have been reported in simple free viewing conditions where participants process image patterns such as photographs. In neurotypical individuals eye movements cover wide areas, such that they uniformly include salient image features. In participants with schizophrenia, ocular exploration tends to be spatially more limited, with shorter scanpaths as compared to neurotypical participants [37,38,39].

As it has been demonstrated for other attentional mechanisms [40], similar selection/inhibition mechanisms operated by selective attention occur for both the physical (i.e., visual field) and the representational (e.g., memory buffer) space. This is the case also for the center-surround profile of the attentional focus. Recent findings demonstrated that the representation of information in visual working memory is maintained with the same center-surround profile that acts in visuo-spatial selection [41]. This suggests interesting speculations about how an altered general mechanism for selection of information, which we demonstrated in the present study for individuals with SSD, can impact higher-order cognitive functions given its similar action both in the physical and representational space.

Although the precise neurophysiological mechanisms that implement the center-surround profile of the attentional focus are not known, a reasonable hypothesis is that an altered suppression of visual information surrounding the focus of attention can arise from an altered modulation of the top-down effect that fronto-parietal areas of the dorsal attentional network [4,42,43] exert on lower-level visual areas. In favor of a top-down modulation is the evidence showing that the center-surround profile appears only in tasks requiring spatial scrutiny, while is not elicited when stimulus discrimination can be achieved without precise spatial localization [19,20,23]. Moreover, a study by Boehler and collaborators [44] reported MEG findings showing that surround suppression appears in neurotypical adults with a timing (i.e., >175 ms) that is well beyond the time taken to complete the initial feedforward sweep of processing in the visual system. Although with a simplified paradigm, ERPs results that are compatible with a relatively late modulation (i.e., N2 ERP) have been reported also in children [22].

Recent evidence in both primate and human electrophysiology highlighted the potential role of alpha/low-beta oscillations in fronto-parietal areas and of the connectivity between these regions and visual areas [45,46,47] as a potential mechanism serving the function of reducing low-level sensory crowding and uncertainty [48,49,50,51,52]. There is growing awareness of the importance of brain oscillations as a means to better understand neurodevelopmental and psychiatric disorders, such as SSD and autism spectrum disorders (ASD). In particular, brain oscillations would provide an important method to understand how top-down signaling influences bottom-up input. Recently, starting from this point of view, Tarasi et al. [53] proposed a theoretical framework according to which disturbances in the oscillatory profile in both SSD and ASD individuals might hamper the appropriate trade-off between descending top-down (and predictive) signaling and ascending bottom-up sampling of sensory information. In particular, top-down and predictive signals would be overweighted in SSD and potentially underweighted in ASD. This prediction is in line with the evidence reported here for individuals with SSD, which is compatible with the idea that top-down influences are exerting an excessive modulation on lower level visual processing. Moreover, that prediction is also congruent with the evidence previously reported by Ronconi et al. [22], who investigated the center-surround profile of the attentional focus in children and adolescents with ASD. Two independent experiments performed by these authors provided psychophysical and electrophysiological evidence of a weaker suppression in spatial positions surrounding the focus of attention, suggesting effectively an underweighted top-down influence on lower-level visual processing.

While for ASD the pattern emerging when analyzing the extent of their inhibitory ring seems opposite to what has been found here for individuals with SSD, there are other conditions which have been associated, similarly to SSD, with an exaggerated attentional bottleneck. Indeed, recent findings from multiple experimental approaches converge in showing that psychopathic individuals would struggle to process multiple streams of information simultaneously [54,55]. This peculiar attentional pattern emerges in different types of attentional paradigms, such as the attentional blink for which psychopathic individuals show a reduced susceptibility [56]; it also emerges in dual tasks where psychopaths exhibit an exaggerated bottleneck which produces marked and long-lasting interference in reaction times and a reduction in related ERPs indices [57]. An exaggerated attention bottleneck not only may induce psychopathic individuals to be more effective at filtering out distractions and focusing on personal goals, but also lead them to over-prioritize goal-relevant or salient information at the expense of other important context-relevant information (e.g., face emotional cues), leading to degraded and fractioned representations that can impact on the choice of appropriate behavioral responses [55]. To further corroborate the predictions of the attentional bottleneck theory of psychopathy, future research might consider testing the inhibitory ring of the attentional focus in this population using experimental paradigms such as the one employed in the present study.

Although the study has produced convincing results, some limitations must be acknowledged. Firstly, there is large variability in the profiles of SSD patients, which could limit the generalization of the results. In general, it is hard to obtain homogeneous samples of participants since the disease is multifaceted and its symptoms may increase (or fluctuate) in time. A study involving several institutions would help to address this limitation. In order to collect enough data, the present study has required three years. Future studies are expected to extend the present results by evaluating whether different diagnoses show a similar pattern. We hypothesize that these results will be replicated, since the attentional behavior described here is likely to be directly dependent on a neurocognitive mechanism that is at the core of the disorder. A second limitation depends on the attentional requirement of our task. As in many studies, it was possible to test SSD participants only when their clinical condition was optimal. Therefore, while it is not possible to address this constraint, it may mask some effects and hinder the possibility of describing the mechanism underlying the regulation of the attentional focus in other more impairing conditions.

The present study was conducted in patients with SSD undergoing antipsychotic medications, as most of the studies conducted in this clinical population. This stimulates a consideration regarding the possible influence that medication had on the results observed here. Few reasons lead to exclude a role for medication in determining the peculiar pattern of visual attention delineated here. Firstly, patients performed the task with an accuracy that, overall, was largely above chance and only slightly below the accuracy level observed in the control group. Thus, all patients included could perform the task at a level which was adequate to measure specific changes in the nature of attentional suppression measured by our task. Moreover, for each participant, the main data analysis was performed by adjusting the accuracy values that were measured in the critical (probe) condition, to accuracy values that were observed in the baseline (target) condition. This should have further ensured that any general impairment in attention (e.g., poor sustained attention) did not affect the spatial profile of the attentional focus measured in the SSD group. Secondly, the patients involved in this study were all taking an intermediate dosage of second-generation antipsychotic drugs. All of them already reached stable clinical symptomatology and none of them were hospitalized at the time of the study. While interindividual differences were controlled, the problem of a different behavior with respect to non-medicated patients is still unresolved. However, as a final relevant consideration, the few studies that examined the effect of antipsychotic treatment in drug-naïve patients with schizophrenia found that there are no evident effects of antipsychotic treatment on cognition (for a review see [58]). Coherently, a meta-analysis from Fatouros-Bergman et al. [58] investigated cognitive performance in studies testing only antipsychotic drug-naïve patients, confirming results observed in previous meta-analyses conducted in patients undergoing antipsychotic treatments [59]; specifically, also for drug-naïve patients with schizophrenia, the authors observed deficits in several areas of cognition (i.e., verbal, visual and working memory, speed of processing, attention, executive functions) with medium to large effect sizes.

To conclude, the present study shows that attentional deficits in schizophrenic patients may be based on their tendency to excessively suppress sensory inputs surrounding the attentional focus. This finding is likely to influence both models about schizophrenia and the clinical practice. The former should include a primary role for inhibition, which is expected to be responsible for the fragmented perception of reality. The clinical practice should consider that interaction and cognitive functioning may improve if the stimulation is provided centrally with respect to the focus, for example avoiding communication when SSD patients are not attentive to the source. Alternatively, strategies for enhancing elaboration of information that are outside the focus of attention should be pursued. We suggest future work to validate the ideas described in the present study, for example extending the investigation to other sensory modalities or to higher order cognitive functions (e.g., working memory), so that a better understanding of the SSD functioning could be achieved.

## Figures and Tables

**Figure 1 brainsci-13-00211-f001:**
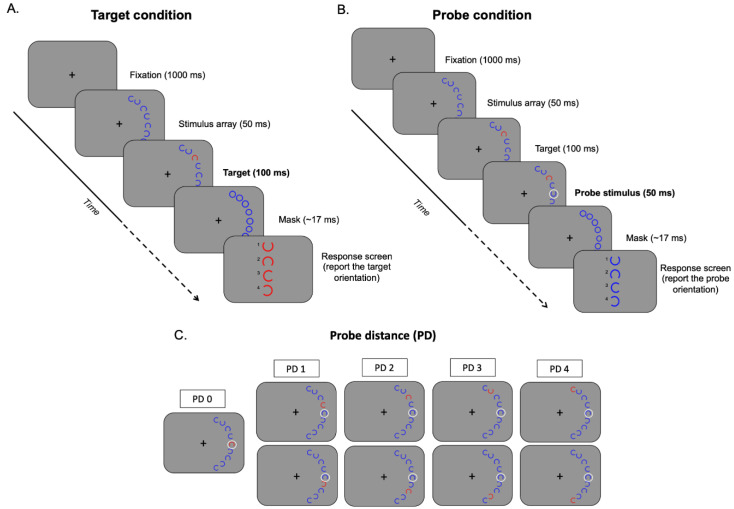
Schematic representation of the task procedure used to measure the spatial profile of the attentional focus in SSD. (**A**) Target condition. (**B**) Probe condition. (**C**) Determination of the different probe distances (PD) in the probe condition, where PD0 corresponds to the center of the attentional focus, while PD1–PD4 represent positions progressively farther from the center of the focus.

**Figure 2 brainsci-13-00211-f002:**
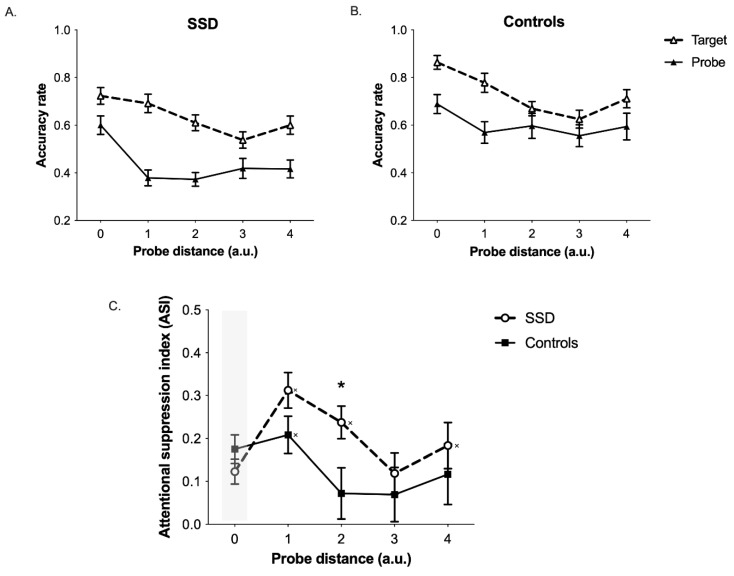
Results showing the spatial profile of the attentional focus in SSD. (**A**,**B**) Plots showing mean accuracies in the (**A**) SSD and (**B**) control group as a function of task condition and probe distances (PD) (PD0 = center of the attentional focus. PD1–PD4 = positions progressively farther from the center of the focus). Chance level = 0.25. (**C**) Attentional suppression index (ASI), computed as the accuracy difference between the probe and the target conditions, as a function of group (SSD and controls) and PD. The shaded area represents data points not considered for planned comparisons. * = *p* < 0.05 for independent samples *t*-tests (SSD vs. controls); × = *p* < 0.05 for within-samples *t*-tests against 0. Bars represent the SEM.

**Table 1 brainsci-13-00211-t001:** Details of the antipsychotic medications undergoing at the time of the experiment in participants with schizophrenia (*N* = 24).

Drug	Number of Patients	Average Daily Dosage
Clozapine	7	270 mg
Olanzapine	6	10 mg
Aripiprazole	6	10 mg
Paliperidone	3	5 mg
Fluphenazine	1	1 mg
Quetiapine	1	100 mg

## Data Availability

Data are available upon requests to the authors.

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
