# Peer review of "Wider and Stronger Inhibitory Ring of the Attentional Focus in Schizophrenia"

_brainsci, 2023, doi:10.3390/brainsci13020211_

Round 1

Reviewer 1 Report

In the present study, semicircular search arrays composed of blue Landolt Cs were displayed to the left or right of fixation, one of which turned red after a brief interval. In one (baseline) condition, subjects had to report the orientation of the red C. In another condition (probe), the C turning red was followed by the onset of a white circle surrounding one of the blue Cs, in which case subjects had to report the orientation of the probed blue C. The manipulation of interest in the present experimental context was that concerning the spatial distance between the red C in the baseline condition and the position of the probed blue C in the probe condition, which was sistematically varied by always displaying the probed blue C in either positions on the horizontal midline intersecting central fixation (i.e., at 3 or 9 o'clock) and the red C in one of the other positions (adjacent, PD1, and or farther away, PD2 to PD4) in the search array. The exception was represented by cases in which the red C and the probed blue C were both displayed, in distinct trials, at 3 or 9 o'clock positions. The results showed that probed blue Cs displayed in the surroundings of the position occupied by the red C were reported less accurately relative to more distant positions, the more so when controls were compared to patients affected by schizophrenia.

I must say I liked reading this nifty and well-written paper, which clearly has the potential to make a significant contribution to the vast scientific literature revolving around human attention and its pathological flaws. The paradigm used is state-of-the-art, and the results particularly solid. In the editor's shoes, I would bat strongly for this paper to be in print after a few, lame, modifications, which I list in points below.

1. It would be nice if the authors could elaborate slighly more in details some peculiar characteristics of the results before jumping stright into the Discussion, which is at present mostly occupied by considerations that appear to be disconnected from the nice empirical picture graphed in Fig. 2. It is not immediately obvious what is the match between the results predicted and observed. Nor is it immediately obvious why, at PD0, accuracy for probe and target tend to diverge and why at PD4 there is no recovery from the suppressive ring effect in normals... In general there is a lot of info to expand upon to meet my request.

2. Part of the discussion is dedicated to bridge the present results with the potential role of top-down modulations, which are perahaps more critical when search is not as easy as in the present case, where -- I assume -- singling out targets and probes can be carried based on feed-forward volleys of activation. There are papers that can be cited in support.

3. The discussion around the different modes of attentional selection in normals vs patients affected by schizofrenia reminded me of a paper I found quite intriguing that the authors may want to consider to widen the range of peculiarities associated with the hypotesis of a restricted bandwidth of input selection in pathology in general. The paper by Baskin-Sommers and Brazil (TICS, 2022) is attached.

Minor stuff:

The paper needs rereading for the few typos I stumbled on while reading.

Would the authors be willing to reconsider the legends of Fig. 2 by changing 'baseline' with target and leaving 'probe' for probe... It seems just more coherent conceptually (plus helping the read out, in my view).

Congratulations for this (demanding... years to collect data!) nice report.

RDA

Author Response

REVIEWER 1 (R1)

In the present study, semicircular search arrays composed of blue Landolt Cs were displayed to the left or right of fixation, one of which turned red after a brief interval. In one (baseline) condition, subjects had to report the orientation of the red C. In another condition (probe), the C turning red was followed by the onset of a white circle surrounding one of the blue Cs, in which case subjects had to report the orientation of the probed blue C. The manipulation of interest in the present experimental context was that concerning the spatial distance between the red C in the baseline condition and the position of the probed blue C in the probe condition, which was sistematically varied by always displaying the probed blue C in either positions on the horizontal midline intersecting central fixation (i.e., at 3 or 9 o'clock) and the red C in one of the other positions (adjacent, PD1, and or farther away, PD2 to PD4) in the search array. The exception was represented by cases in which the red C and the probed blue C were both displayed, in distinct trials, at 3 or 9 o'clock positions. The results showed that probed blue Cs displayed in the surroundings of the position occupied by the red C were reported less accurately relative to more distant positions, the more so when controls were compared to patients affected by schizophrenia.

I must say I liked reading this nifty and well-written paper, which clearly has the potential to make a significant contribution to the vast scientific literature revolving around human attention and its pathological flaws. The paradigm used is state-of-the-art, and the results particularly solid. In the editor's shoes, I would bat strongly for this paper to be in print after a few, lame, modifications, which I list in points below.

Authors: We would like to thank the Reviewer for the positive evaluation of our manuscript and for the important recommendations on how to improve it. We have done our best to address his suggestions as detailed below.  

R1: 1. It would be nice if the authors could elaborate slighly more in details some peculiar characteristics of the results before jumping stright into the Discussion, which is at present mostly occupied by considerations that appear to be disconnected from the nice empirical picture graphed in Fig. 2. It is not immediately obvious what is the match between the results predicted and observed. Nor is it immediately obvious why, at PD0, accuracy for probe and target tend to diverge and why at PD4 there is no recovery from the suppressive ring effect in normals... In general there is a lot of info to expand upon to meet my request.

Authors: We agree that it was not so clear the link between the hypotheses and the observed results. In the second paragraph of the Discussion we have substantially extended the description of the central results of the paper, providing a more in depth analysis of why they agree with the hyperfocusing hypothesis of SSD. On page 7 we now write:

As compared to neurotypical controls, individuals with SSD show stronger filtering of visual information for spatial positions outside their focus of attention. Moreover, they show a wider spatial extension of the inhibitory annulus surrounding the focus of attention. Thus, results observed in the SSD group matched substantially with the hypothesis of an hyperfocusing of attention in SSD, which has been proposed by some previous studies [15,16]. Hyperfocusing, indeed, should lead to an increased filtering of information falling outside the attentional focus, both in terms of filtering strength and in terms size of the inhibitory areas. Our findings confirmed both these predictions, showing that the filtering of visual information outside the focus of attention is more accentuated and shows a wider spatial extension in individuals with SSD than in neurotypical individuals. This pattern is particularly evident when looking at the attentional suppression index (ASI) at PD2 in the two groups: indeed, individuals with SSD in this position showed a stronger ASI as compared to controls. Moreover, in this same spatial position, there was a complete recovery from suppression in controls: indeed, the ASI was not significantly different from 0 at PD 2, 3 and 4 in the control group, supporting the claim that the suppressive ring effect was significant only at PD1. Contrarily, the attentional suppression was still evident in the SSD group at PD2 and at PD4.

On a marginal note, regarding the recovery from suppression at PD4, we agree that the trend in Figure 2C would be expected to be a bit different; we would like to point out, at the same time, that in controls the ASI was not significantly different from 0 at PD 2, 3 and 4, supporting the claim that the suppressive ring effect was significant only at PD1.

Finally, regarding the differences between probe- and target- orientation discrimination accuracy observed at PD0, which was observed independently from the group, we might speculate it arises potentially by an increased cognitive effort required by the dual task. We did not directly test this specific difference and did not include a Discussion of this aspect as we feel that, although interesting, it is outside the scope of the article.

R1: 2. Part of the discussion is dedicated to bridge the present results with the potential role of top-down modulations, which are perahaps more critical when search is not as easy as in the present case, where -- I assume -- singling out targets and probes can be carried based on feed-forward volleys of activation. There are papers that can be cited in support.

Authors: This is an interesting point, and we thank the Reviewer for pointing it out as we realized only after the revision that this aspect was not properly discuss in the manuscript.

We have integrated the Discussion section with the following part on page 8, to better explain why we and other authors consider that the inhibitory ring arises as a result of a feedback loop, and not a feedforward sweep of activation. 

In favor of a top-down modulation is the evidence showing that the center-surround profile appears only in tasks requiring spatial scrutiny, while is not elicited when stimulus discrimination can be achieved without precise spatial localization [19, 20, 23]. Moreover, a study by Boehler and collaborators [44] reported MEG findings showing that surround suppression appears in neurotypical adults with a timing (i.e. >175 ms) that is well beyond the time taken to complete the initial feedforward sweep of processing in the visual system. Although with a simplified paradigm, ERPs results compatible with a relatively late modulation (i.e. N2 ERP component) have been reported also in children [22].

R1: 3. The discussion around the different modes of attentional selection in normals vs patients affected by schizofrenia reminded me of a paper I found quite intriguing that the authors may want to consider to widen the range of peculiarities associated with the hypotesis of a restricted bandwidth of input selection in pathology in general. The paper by Baskin-Sommers and Brazil (TICS, 2022) is attached.

Authors: Thank for the relevant suggestion. It has been a very interesting reading, indeed. We have widened the Discussion section including a link to this peculiar attentional bottleneck in psychopathy on pages 8-9:

While for ASD the pattern emerging when analyzing the extent of their inhibitory ring seems opposite to what has been found here for individuals with SSD, there are other conditions which have been associated, similarly to SSD, with an exaggerated attentional bottleneck. Recent findings from multiple experimental approach, indeed, converge in showing that psychopathic individuals would struggle to process multiple streams of information simultaneously [54, 55]. This peculiar attentional pattern emerges in different types of attentional paradigms, such as the attentional blink for which psychopathic individuals show a reduced susceptibility [56]; it also emerges in dual tasks where psychopaths exhibit an exaggerated bottleneck which produces marked and long-lasting interference in reaction times and a reduction in related ERPs indices [57]. An exaggerated attention bottleneck not only may induce psychopathic individuals to be more effective at filtering out distractions and focusing on personal goals, but also lead them to over-prioritize goal-relevant or salient information at the expense of other important context-relevant information (e.g. face emotional cues), leading to degraded and fractioned representations that can impact on the choice of appropriate behavioral responses [55]. To further corroborate the predictions of the attentional bottleneck theory of psychopathy, future research might consider testing the inhibitory ring of the attentional focus in this population using experimental paradigms like the one employed in the present study.

R1: Minor stuff:

R1: The paper needs rereading for the few typos I stumbled on while reading.

Authors: The paper has been extensively checked for typos. If needed, the paper could undergo a check from a Native English speaker: in this case, however, the check will be postponed after having addressed properly all the content issues of the manuscript.

R1: Would the authors be willing to reconsider the legends of Fig. 2 by changing 'baseline' with target and leaving 'probe' for probe... It seems just more coherent conceptually (plus helping the read out, in my view).

Authors: Thank for this suggestion. We have revised the labels for our experimental conditions accordingly throughout the text and figures.

Reviewer 2 Report

I read the study of Ronconi et al entitled “ WIDER AND STRONGER INHIBITORY RING OF THE ATTENTIONAL FOCUS IN SCHIZOPHRENIA “, which aims to evaluate the spatial profile of the attentional focus in individuals with SSD.

1- Please adapt the language of the article to an easier way to be more pleasant to the reader, especially in the abstract

2- Please reduce the introduction and make it more relevant to the matter of the study

Author Response

REVIEWER 2 (R2)

I read the study of Ronconi et al entitled “ WIDER AND STRONGER INHIBITORY RING OF THE ATTENTIONAL FOCUS IN SCHIZOPHRENIA “, which aims to evaluate the spatial profile of the attentional focus in individuals with SSD.

R2: 1- Please adapt the language of the article to an easier way to be more pleasant to the reader, especially in the abstract

Authors: Thank for the suggestion. We have revised the abstract to help its readability. However, if further changes will be required, we will submit the manuscript to a professional service. Please check the answer to R1.

R2: 2- Please reduce the introduction and make it more relevant to the matter of the study

Authors: Thank for the suggestion. We have shortened the first part of the Introduction so that the flow is now more focused on the literature related to this study.

Reviewer 3 Report

This paper delivers experimental evidence in contradiction to conceptual views inherited from the Gestalt-psychopathology, namely that patients with schizophrenia tend to focus their attention more on irrelevant stimuli (generated from the surrounding background), then a current object (or central figure) and replace one with the other.

According to the reported results from this study patients actually have wider inhibition or filtering of signals outside the current attentional focus. Authors claim this finding as possible explanatory mechanism behind attentional deficit in schizophrenia.

Therefore certain concerns need to be addressed.

1. Authors should introduce a clear and common definition for "neurotypical" control as well as to set a definition for the broader term of the schizophrenia spectrum disorder in the context of this particular study.

2. Authors should add a table of the antipsychotic medications, received by the patients with average doses for the different classes. This is potential confound of the study as the observed effects may well be regarded in terms of therapeutic response rather then as mechanism of disorder.

Author Response

REVIEWER 3 (R3)

This paper delivers experimental evidence in contradiction to conceptual views inherited from the Gestalt-psychopathology, namely that patients with schizophrenia tend to focus their attention more on irrelevant stimuli (generated from the surrounding background), then a current object (or central figure) and replace one with the other.

According to the reported results from this study patients actually have wider inhibition or filtering of signals outside the current attentional focus. Authors claim this finding as possible explanatory mechanism behind attentional deficit in schizophrenia.

Therefore certain concerns need to be addressed.

R3: 1. Authors should introduce a clear and common definition for "neurotypical" control as well as to set a definition for the broader term of the schizophrenia spectrum disorder in the context of this particular study.

Authors: Thank you for these important suggestions. In the revised text we have tried to define better, in the ‘Participants’ section, why we refer to our control participant with the term ‘neurotypical’. We have also specified that, although we have used SSD to label our group, all patients included in the study received a diagnosis of schizophrenia.

On page 3 lines 141-143 we now write:

“Participants of the control group were sampled from the same geographical area and were defined as ‘neurotypical’ since they did not have prior history of any neurological and/or psychiatric disorders.”

On page 3 lines 129-130 we write:

“All patients involved in the present study had a clinical diagnosis of schizophrenia and were recruited at the Psychiatric Service of the Health District of Bozen, Bozen, Italy.”

Finally, on page 3 lines 138-140 we write:

Although, as specified above, all our patients had a diagnosis of schizophrenia, we will use hereafter the label SSD to refer to our group of patients to better adhere to the current standards in the scientific literature.”

R3: 2. Authors should add a table of the antipsychotic medications, received by the patients with average doses for the different classes. This is potential confound of the study as the observed effects may well be regarded in terms of therapeutic response rather then as mechanism of disorder.

Authors: We agree that is important to report the ongoing pharmacological treatments for our patients. We have added such information on Table 1.

Round 2

Reviewer 3 Report

Dear Authors,

thank you for your revisions in line with the peer review report. You are now invited to add a comment in the discussion or limitations (250-350 words) on the influence of medication on your findings. More specifically to what extent your findings might be regarded as  mechanism of disorder rather then as effect of medication. Relevant independent literature should be cited. This will foster your conclusion.

Reviewer 3

Author Response

Reviewer 3 (Round 2):

Dear Authors,

thank you for your revisions in line with the peer review report. You are now invited to add a comment in the discussion or limitations (250-350 words) on the influence of medication on your findings. More specifically to what extent your findings might be regarded as mechanism of disorder rather then as effect of medication. Relevant independent literature should be cited. This will foster your conclusion.

Authors: We would like to thank the Reviewer for this further suggestion on how to improve our discussion. In the newly revised version we have included a new paragraph that deeply discuss the potential influence of medication on the reported findings. On page 9 we included the following paragraph:

“The present study was conducted in patients with SSD undergoing antipsychotic medications, as most of the studies conducted in this clinical population. This stimulates a consideration regarding the possible influence that medication had on the results observed here. Few reasons lead to exclude a role for medication in determining the peculiar pattern of visual attention delineated here. Firstly, patients performed the task with an accuracy that, overall, was largely above chance and only slightly below the accuracy level observed in the control group. Thus, all patients included could perform the task at a level which was adequate to measure specific changes in the nature of attentional suppression measured by our task. Moreover, the main data analysis was performed by adjusting, for each participant, the accuracy values measured in the critical (probe) condition to accuracy values observed in the baseline (target) condition. This should have further ensured that any general impairment in attention (e.g. poor sustained attention) did not affect the spatial profile of the attentional focus measured in the SSD group. Secondly, the patients involved in this study were all taking an intermediate dosage of second-generation antipsychotic drugs. All of them already reached stable clinical symptomatology and none of them was hospitalized at the time of the study. While interindividual differences were controlled, the problem of a different behavior with respect to non-medicated patients is still unresolved. However, as a final relevant consideration, the few studies that examined the effect of antipsychotic treatment in drug-naïve patients with schizophrenia found that there are no evident effects of antipsychotic treatment on cognition (for a review see [58]). Coherently, a meta-analysis from Fatouros-Bergman et al. [58] investigated cognitive performance in studies testing only antipsychotic drug-naïve patients, confirming results observed in previous meta-analyses conducted in patients undergoing antipsychotic treatments [59]; specifically, also for drug-naïve patients with  schizophrenia the authors observed deficits in several areas of cognition (i.e. verbal, visual and working memory, speed of processing, attention, executive functions) with medium to large effect sizes.”